# Fault Diagnosis of Planetary Gearbox Based on Dynamic Simulation and Partial Transfer Learning

**DOI:** 10.3390/biomimetics8040361

**Published:** 2023-08-12

**Authors:** Mengmeng Song, Zicheng Xiong, Jianhua Zhong, Shungen Xiao, Jihua Ren

**Affiliations:** 1College of Information, Mechanical and Electrical Engineering, Ningde Normal University, Ningde 352000, China; t1119@ndnu.edu.cn; 2College of Mechanical Engineering and Automation, Fuzhou University, Fuzhou 350202, China; 3Dongguan Xinghuo Gear Co., Ltd., Dongguan 523000, China; sharyren@xhgear.com

**Keywords:** planetary gearbox, fault diagnosis, dynamics simulation, partial transfer learning

## Abstract

To address the problem of insufficient real-world data on planetary gearboxes, which makes it difficult to diagnose faults using deep learning methods, it is possible to obtain sufficient simulation fault data through dynamic simulation models and then reduce the difference between simulation data and real data using transfer learning methods, thereby applying diagnostic knowledge from simulation data to real planetary gearboxes. However, the label space of real data may be a subset of the label space of simulation data. In this case, existing transfer learning methods are susceptible to interference from outlier label spaces in simulation data, resulting in mismatching. To address this issue, this paper introduces multiple domain classifiers and a weighted learning scheme on the basis of existing domain adversarial transfer learning methods to evaluate the transferability of simulation data and adaptively measure their contribution to label predictor and domain classifiers, filter the interference of unrelated categories of simulation data, and achieve accurate matching of real data. Finally, partial transfer experiments are conducted to verify the effectiveness of the proposed method, and the experimental results show that the diagnostic accuracy of this method is higher than existing transfer learning methods.

## 1. Introduction

A planetary gearbox (PG) is a key component of rotating machinery. Due to its advantages of large bearing capacity, small volume, and high transmission efficiency, it has been widely applied in mechanical transmission systems in industries such as wind power, aviation, lifting, and transportation. However, PG often works in harsh environments, making it susceptible to malfunctions. If PG fails, it may cause the entire transmission system to degrade and fail, even causing catastrophic damage and huge economic losses [1]. Therefore, researching fault diagnosis methods for PG has important practical significance for ensuring stable operation and prolonging the service life of mechanical equipment [2]. By conducting relevant research and establishing reliable fault diagnosis models, people can detect problems in a timely manner and take corresponding measures. This can not only reduce the maintenance costs of mechanical equipment but also avoid larger faults and losses.

When using machine learning and deep learning methods for PG fault diagnosis, huge quantities of labeled fault data are usually necessary. However, for actual industrial production, most data is collected while the machine is running normally [3], making it challenging to obtain vast and detailed fault data. To address this issue, an abundance of simulated fault data can be gathered through dynamic simulation analysis, and then transfer learning (TL) methods can be used to narrow the disparity between simulated and real data. Thus, the diagnostic information in the simulated data is able to be applied to the fault diagnosis of real PG. Li et al. [4] simulated vibration signals using a lumped parameter dynamic model and then used a CNN-based TL network to obtain domain-invariant features from various domains in order to classify faults. Dong et al. [5] generated an abundance of simulated data using dynamic models and then used CNN and parameter transfer methods to apply the learned fault diagnosis knowledge to practical scenarios, solving the problem of small samples. Li et al. [6] trained a deep neural network model using computer-simulated data to deal with the challenge caused by an insufficient amount of labeled fault data and used TL to narrow the discrepancy between the simulated and actual domains. Zhu et al. [7] introduced a defect vibration model for simulating fault vibration signals and used real and simulated signals as the target domain and source domain of TL fault diagnosis methods, demonstrating the method’s effectiveness and superiority through experimentation. Liu et al. [8] generated simulated vibration signals using a phenomenological model and then used domain adversarial neural networks to train adversarial data among source and target domains. According to the experimental findings, this means can produce excellent classification accuracy with just relatively little real data.

Diagnostic methods based on dynamic simulation and TL usually presume that the label space of simulation data and real data are identical. But when this method is applied to real planetary gear fault diagnosis, simulation data can contain all possible fault categories while the real planetary gearbox may only have one or a few faults, which means that the label space of real data is a subset of the label space of simulation data. This can lead to interference from outlier label space in simulation data, causing mismatching. Partial transfer learning (PTL) methods can help reduce mismatching. Wang et al. [9] proposed a balanced adversarial domain adaptive network for fault diagnosis tasks in partial transfer scenarios, which alleviated the mismatching problem by introducing balancing strategies and class-level weights. Li et al. [10] proposed a class-weighted adversarial neural network that encourages positive transfers of shared classes and ignores source class outliers through class-weighting strategies. Sun et al. [11] suggested a game theory-enhanced domain adaptation network to solve partial domain adaptation problems. The network constructs three attention matrices using maximum mean discrepancy, Jensen-Shannon divergence, and Wasserstein distance and generates the best probability weight through the combination of game theory weights, thereby filtering out irrelevant source domain samples and improving mechanical fault diagnosis performance. Li et al. [12] suggested a new weighted adversarial transfer network that filters out irrelevant source domain samples and improves the performance of the target task through weighted learning. Jiao et al. [13] proposed a domain adaptive network based on classifier inconsistency. The network uses two discriminative 1D-CNN as the basic architecture and promotes active network training by identifying and emphasizing source domain samples with the same classification as the target domain. At the same time, the classifier inconsistency is added in order to direct the model towards learning discriminative and domain-invariant representations for precise classification of unlabeled target data. Kuang et al. [14] proposed a two-stage double-weight consistency-induced partial domain adaptive network. This network obtains double-level composite weights from class-level and sample-level weights through double-weight consistency-induced weighting strategies, enabling selective mapping of source diagnosis knowledge to the target domain.

To deal with the challenge of scarce labeled fault data in real PG fault diagnosis, this paper establishes a dynamic simulation model of PG to obtain abundant fault simulation data. But simulation data and actual data are distinct, and the label space is also heterogeneous. To solve these problems, this paper introduces multiple domain classifiers and weighted learning schemes on the basis of existing domain adversarial TL methods, evaluates the transferability of simulation data, adaptively measures their contributions to label predictor and domain classifiers, filters out the interference of irrelevant categories of simulation data, achieves accurate matching of real data, and thus improves the diagnostic accuracy of transfer tasks.

The main contributions of this paper are as follows:Through the rigid-flexible coupling dynamic model of PG, a wealth of fault simulation data is obtained, and then the problem of scarcity of labeled fault data in real-world scenarios is solved.By introducing multiple domain discriminators and a weighted learning scheme, the interference from simulation data of irrelevant categories is filtered, thereby improving the diagnostic accuracy of partial transfer tasks.

The remainder of this paper is arranged as below: Section 2 introduces the relevant theories. Section 3 describes the proposed means. Section 4 studies a practical case. Section 5 summarizes this paper.

## 2. Theoretical Background

### 2.1. Partial Transfer Learning

In TL, it is often assumed that the label space of the source domain (*D_s_*) samples, *C_s_*, is the same as that of the label space of the target domain (*D_t_*) samples, *C_t_*. However, in real-world applications, *C_t_* is more likely to be a subset of *C_s_*. In this case, all labels in *D_t_* are shared by both *D_s_* and *D_t_*, and *C_t_* is also known as the shared label space. There are also some labels in *D_s_* that are unique to it, known as the outlier label space (*C_s_*-*C_t_*), which can lead to mismatches between *D_s_* and *D_t_* samples and affect the accuracy of the transfer task. PTL aims to promote the positive transfer of samples in the shared label space while suppressing the negative transfer of samples in the outlier label space when *C_t_* is a subset of *C_s_*, thereby improving the accuracy of the transfer task [15]. Figure 1 illustrates the concept behind traditional TL and PTL. In the figure, *D_s_* samples have three types of labels: △, ○, and □, while *D_t_* samples only have two types of labels: △ and □. In this case, ○ in *D_s_* is an outlier label, which may lead to mismatching with *D_t_* samples during TL. However, PTL methods can recognize and filter out outlier labels in *D_s_*, effectively reducing the risk of mismatching and improving model performance.

### 2.2. Residual Neural Network

For deep neural networks, the number of layers is crucial. The deeper the network, the richer its ability to extract hierarchical features and its recognition and classification capabilities are also enhanced. However, for traditional CNN, too many layers can cause gradient vanishing or explosion, making the network difficult to train. To address the issue, He et al. [16] proposed Residual Neural Networks (ResNet) in 2015. ResNet usually includes convolutional layers, pooling layers, residual blocks, and fully connected layers. Figure 2 illustrates a schematic diagram of residual blocks, where *x* is the input, *H*(*x*) = *F*(*x*) + *x* is the output, and *F*(*x*) = *H*(*x*) − *x* is the residual. The residual block has two branches, the residual branch, and the identity mapping branch. The residual branch consists of two convolutional layers, which are used to fit the residual *F*(*x*), while the identity mapping branch keeps the input *x* unchanged. The output *H*(*x*) of the residual block is obtained by element-wise addition of the two branches and then passed through the ReLu activation function. The introduction of identity mapping ensures that the performance of deep networks is not worse than that of shallow networks, and no additional parameters or computation complexity are added.

### 2.3. Domain Adversarial Neural Network

Domain Adversarial Neural Networks (DANNs) have been extensively implemented in TL. Through the adversarial learning process, the network is capable of extracting domain-invariant features from both *D_s_* and *D_t_*. The adversarial learning process is able to be seen as a two-player game, with the first player being a domain classifier *G_d_* taught to differentiate between *D_s_* and *D_t_* features, and the second player is a feature extractor *G_f_* trained to confuse *G_d_*. The framework of DANN is shown in Figure 3.

To obtain domain-invariant features, during the training process of DANN, the parameters *θ_f_* of *G_f_* are learned by maximizing the loss of the *G_d_*. The parameters *θ_d_* of *G_d_* are learned by minimizing its loss. In addition, minimizing the loss of the label predictor *G_y_* ensures a low *D_s_* classification error. The overall loss function of DANN is shown in Equation (1) [17]:(1)$$\begin{array}{ll} L\left(\theta_{f},\theta_{y},\theta_{d}\right) & =\frac{1}{n_{s}}\sum_{x_{i}\in D_{s}}L_{y}\left(G_{y}\left(G_{f}\left(x_{i}\right)\right),y_{i}\right)\\ & -\frac{\lambda }{n_{s}+n_{t}}\sum_{x_{i}\in D_{s}\cup D_{t}}L_{d}\left(G_{d}\left(G_{f}\left(x_{i}\right)\right),d_{i}\right) \end{array}$$

In the formula, *L_y_* is the loss function of *G_y_*, *L_d_* is the loss function of *G_d_*, *d_i_* is the domain label of the *i*-th sample, and *λ* is the hyperparameter that balances *L_y_* and *L_d_*. The parameter optimization for DANN is:(2)$$\left({\hat{\theta }}_{f},{\hat{\theta }}_{y}\right)=\arg\underset{\theta_{f},\theta_{y}}{\min}\;L\left(\theta_{f},\theta_{y},\theta_{d}\right)$$
(3)$$\left({\hat{\theta }}_{d}\right)=\arg\underset{\theta_{d}}{\max}\;L\left(\theta_{f},\theta_{y},\theta_{d}\right)$$

## 3. Proposed Method

### Weighted Domain Adversarial Neural Network Diagnostic Model

When *D_t_* label space *C_t_* is a subset of *D_s_* label space *C_s_*, if existing TL fault diagnosis models are used, *D_t_* samples may be incorrectly matched with samples belonging to the outlier label space *C_s_*-*C_t_* in *D_s_*, resulting in reduced diagnostic accuracy. To deal with this problem, this paper suggests a domain adversarial neural network with a weighted learning strategy to promote the positive transfer of the shared label space *C_t_* and suppress the negative transfer of the outlier label space *C_s_*-*C_t_* in *D_s_*. The network framework is shown in Figure 4.

This network includes a feature extractor *G_f_*, a label predictor *G_y_*, and |*C_s_*| domain classifiers *G_d_*. *G_f_* and *G_y_* are composed of ResNet-18. *G_f_* is the feature extraction part of ResNet-18, including convolutional layers, pooling layers, and residual blocks. To extract more effective features, CBAM [18] is added to each residual block. *G_y_* corresponds to the output of ResNet-18 and includes a fully connected layer and a softmax classification layer. |*C_s_*| domain classifiers *G_d_* have the same structure, including three fully connected layers. The *c*-th domain classifier $G_{d}^{c}$ (*c* = 1, 2, …, |*C_s_*|) is responsible for matching the *D_s_* sample with label *c* and the *D_t_* sample with label *c*. Therefore, in $G_{d}^{c}$, samples with label *c* should be assigned larger weights, while samples with other labels should be assigned smaller weights. In addition, only the domain classifier is accountable for matching the shared label space *C_t_* can promote positive transfer, while the domain classifier accountable for matching the outlier label space *C_s_*-*C_t_* will introduce noise. Therefore, it is necessary to reduce the weight of the domain classifiers responsible for matching the outlier label space *C_s_*-*C_t_*.

As the labeled samples in *D_t_* are unknown during model training, it is not possible to determine the weights based on labels. Joint distribution adaptation (*JDA*) [19] is often used to calculate differences between samples, and this paper will calculate weights using *JDA* values. The calculation formula for the weight matrix ***W_d_*** assigned to the domain classifier is shown in Equations (4)–(6):(4)$$j_{i}^{c}=\left\{\begin{array}{c} j_{i}^{sc}:mean\left(\sum_{x_{j}\in D_{s}^{c}}JDA\left(G_{f}\left(x_{j}\right),G_{f}\left(x_{i}\right)\right)\right),x_{i}\in D_{s}\\ j_{i}^{tc}:mean\left(\sum_{x_{j}\in D_{s}^{c}}JDA\left(G_{f}\left(x_{j}\right),G_{f}\left(x_{i}\right)\right)\right),x_{i}\in D_{t} \end{array}\right.$$
(5)$$y_{c}=\frac{1/mean\left(\sum_{i=1}^{n_{t}}j_{i}^{tc}\right)}{\sum_{c=1}^{\left|C_{s}\right|}\left(1/mean\left(\sum_{i=1}^{n_{t}}j_{i}^{tc}\right)\right)}$$
(6)$$W_{d}=[y_{1},y_{2},\ldots ,y_{\left|C_{s}\right|}]$$

In the formula, *mean*(.) calculates the average value, *G_f_*(.) represents the extracted features by the feature extractor, and *JDA*(.) represents the *JDA* value used to measure the difference between the two samples. If the *JDA* value is small, it indicates that the difference between the two samples is small, and there is a significant possibility that they belong to the same label. $j_{i}^{sc}$ represents the discrepancy between the *i*-th *D_s_* sample and *D_s_* sample with label *c*, and $j_{i}^{tc}$ represents the discrepancy between the *i*-th *D_t_* sample and *D_s_* sample with label *c*. By calculating the discrepancy between each *D_t_* sample and *D_s_* sample with label *c*, the probability *y_c_* of label *c* belonging to the shared label space *C_t_* can be obtained. Since the labels in the outlier label space, *C_s_*-*C_t_*, do not belong to *C_t_*, their probabilities *y_c_*, *c*∈*C_s_*-*C_t_* are small enough to reduce the weight of the domain classifiers responsible for *C_s_*-*C_t_*.

The calculation formula for the weight matrix ***W_s_*** assigned to the samples is shown in Equations (7) and (8):(7)$$s_{i}^{c}=\left\{\begin{array}{c} s_{i}^{sc}:\frac{1/j_{i}^{sc}}{\sum_{c=1}^{\left|C_{s}\right|}1/j_{i}^{sc}}\\ s_{i}^{tc}:\frac{1/j_{i}^{tc}}{\sum_{c=1}^{\left|C_{s}\right|}1/j_{i}^{tc}} \end{array}\right.$$
(8)$$W_{s}=\left[\begin{array}{cccc} s_{1}^{1} & s_{1}^{2} & \ldots & s_{1}^{\left|C_{s}\right|}\\ s_{2}^{1} & s_{2}^{2} & \ldots & s_{2}^{\left|C_{s}\right|}\\ \ldots & \ldots & \ldots & \ldots \\ s_{n_{s}+n_{t}}^{1} & s_{n_{s}+n_{t}}^{2} & \ldots & s_{n_{s}+n_{t}}^{\left|C_{s}\right|} \end{array}\right]$$

In the formula, $s_{i}^{sc}$ is the probability that the label of the *i*-th *D_s_* sample is *c*, while $s_{i}^{tc}$ is the probability that the label of the *i*-th *D_t_* sample is *c*.

After incorporating ***W_d_*** and ***W_s_***, the total loss function of the model is outlined below:(9)$$\begin{array}{ll} L\left(\theta_{f},\theta_{y},{\left.\theta_{d}^{c}\right|}_{c=1}^{\left|C_{s}\right|}\right) & =\frac{1}{n_{s}}\sum_{x_{i}\in D_{s}}L_{y}\left(G_{y}\left(G_{f}\left(x_{i}\right)\right),y_{i}\right)\\ & -\frac{\lambda }{n_{s}}\sum_{c=1}^{\left|C_{s}\right|}\left(\sum_{x_{i}\in D_{s}}y_{c}L_{d}^{c}\left(G_{d}^{c}\left(s_{i}^{sc}G_{f}\left(x_{i}\right)\right),d_{i}\right)\right)\\ & -\frac{\lambda }{n_{t}}\sum_{c=1}^{\left|C_{s}\right|}\left(\sum_{x_{i}\in D_{t}}y_{c}L_{d}^{c}\left(G_{d}^{c}\left(s_{i}^{tc}G_{f}\left(x_{i}\right)\right),d_{i}\right)\right) \end{array}$$

In the equation, *θ_f_* represents the parameters of *G_f_*, *θ_y_* represents the parameters of the *G_y_*, $\theta_{d}^{c}$ represents the parameters of $G_{d}^{c}$, *L_y_* represents the loss function of the *G_y_*, $L_{d}^{c}$ represents the loss function of $G_{d}^{c}$, *d_i_* represents the domain label of the *i*-th sample, and *λ* is a hyperparameter that balances *L_y_* and *L_d_*.

The optimization of the model parameters is as follows:(10)$$\left({\hat{\theta }}_{f},{\hat{\theta }}_{y}\right)=\arg\underset{\theta_{f},\theta_{y}}{\min}\;L\left(\theta_{f},\theta_{y},{\left.\theta_{d}^{c}\right|}_{c=1}^{\left|C_{s}\right|}\right)$$
(11)$$\left({\hat{\theta }}_{d}^{1},\ldots ,{\hat{\theta }}_{d}^{\left|C_{s}\right|}\right)=\arg\underset{\theta_{d}^{1},\ldots ,\theta_{d}^{\left|C_{s}\right|}}{\max}\;L\left(\theta_{f},\theta_{y},{\left.\theta_{d}^{c}\right|}_{c=1}^{\left|C_{s}\right|}\right)$$

Compared with a single domain classifier, the multiple domain classifiers used in this paper have two advantages: (1) by using the weight matrix ***W_d_*** of the domain classifiers, the model can emphasize the domain classifiers responsible for the shared label space and suppress the ones responsible for the outlier label spaces, thereby reducing the negative impact of outlier label spaces. (2) The sample weight matrix ***W_s_*** allows *D_t_* samples to only align with *D_s_* samples of one or multiple most relevant labels, thus reducing mismatching.

## 4. Experiment and Analysis

### 4.1. Dataset Comparison and Analysis

The real data for the PG comes from the Drivetrain Diagnostics Simulator (DDS), which is a comprehensive experimental platform for diagnosing power transmission faults. Figure 5 shows the physical model of the DDS experiment platform. During data collection, the variable-speed drive motor has three speeds: 20 Hz, 30 Hz, and 40 Hz, and the magnetic brake has three currents: 0 A, 0.4 A, and 0.8 A (by adjusting the current of the magnetic brake, various loads can be transferred to the output shaft). The sampling frequency is 12,800 Hz.

The simulated data for the PG comes from our previous article, where a rigid-flexible coupled model was established [20]. We used this model to obtain simulation data for four different health conditions of the PG: sun gear broken tooth fault (BR), sun gear crack fault (CR), sun gear tooth missing fault (MI), and normal sun gear (NO). During data acquisition, the input shaft speed was 30 Hz with no load, and the simulation time was 10 s with a simulation step of 128,000. This is equivalent to a working conditions of 30 Hz 0 A for the simulation data, with a sampling frequency of 12,800 Hz. In addition, in our previous article, we also analyzed the effect of the simulation step size on the simulation data, and the results are shown in Figure 6 [20], where Δt represents the time interval between two impacts, *f_m_* represents the meshing frequency, and *f_g_* represents the fault frequency. It can be seen from Figure 6 that the smaller the simulation step size, the more obvious the periodic shock in the time domain diagram and the sideband in the frequency domain diagram, and they are all consistent with the calculated values of the theoretical formula, which verifies the simulation model and simulation data plausibility.

The rigid-flexible coupling model has simplified the PG of the DDS experimental platform to a large extent, and the parameter settings in the model are difficult to completely match with the actual PG. This results in a discrepancy between the simulation and real data, even though the simulated data agrees with the theory. In order to more intuitively demonstrate this difference, this section will analyze and compare the simulation and real data from the perspectives of time-domain, frequency-domain, and probability distribution.

(1) Comparison and analysis of time domain diagrams

Figure 7 shows the time-domain plots of simulated data and real data, both of which were obtained under a 30 Hz 0 A operating condition. It can be observed that there is no clear periodic impulse in either the simulated data or the real data due to a lack of sufficiently high sampling frequency. However, the real data exhibits amplitude modulation, while the simulated data does not. This is because the real data was collected using a fixed-position vibration sensor, while in PG, the planetary gear rotates, causing the distance between the planetary gear and sensor to change. When the planetary gear is closer to the sensor, the measured meshing vibration is larger, and when it is farther away, it is smaller, resulting in amplitude modulation in the time-domain waveform. In contrast, the simulated data measures the angular acceleration of the planetary carrier, which is less affected by the position of the planetary gear and hence does not exhibit amplitude modulation.

(2) Comparison and analysis of frequency domain diagrams

Figure 8 shows the frequency domain comparison of the simulated and real data, both of which were obtained under the same working conditions of 30 Hz 0 A. The frequency domain spectra of both simulated and real data exhibit obvious meshing frequencies and their low harmonics, indicating similar vibration characteristics between the two groups of data. However, the real data does not show higher harmonics of the meshing frequency, which may be due to environmental noise and other factors affecting the high-frequency components of the frequency domain spectra during data acquisition.

(3) Comparison and analysis of probability distribution

From the simulated data with a frequency of 30 Hz and a current of 0 A, as well as the real data from different conditions (30 Hz 0 A, 20 Hz 0.4 A, 40 Hz 0.8 A), 12,800 data points were selected and normalized to [−1,1], resulting in the probability distribution curves shown in Figure 9. Figure 9 illustrates that the probability distribution curve of the simulated data is more concentrated and has a higher peak compared to the real data. This indicates that the probability distribution of the simulated data and the real data differ significantly. In addition, the probability distribution curves of the real data from different conditions are relatively similar, indicating that the difficulty of transfer between the simulated and real data is higher than the difficulty of transfer between real data from various working condition.

### 4.2. Dataset Description

By using overlapping sampling, 512-length data segments were extracted with a resampling step of 50 from both simulated and real data. Then, Short-Time Fourier Transform (STFT) was applied to convert these segments into time-frequency images with a size of 96 pixels in both width and height. There are 2400 time-frequency images for each health condition of simulated and real data, of which 2000 are utilized for training, and the remaining 400 are utilized for testing. Figure 10 shows the time-frequency images of simulated data. After obtaining the time-frequency images, some partial transfer tasks were designed, as shown in Table 1 and Table 2. In Table 1, the real data of 30 Hz 0 A is used as *D_s_*, while the real data of other working conditions are used as *D_t_*. In Table 2, the simulated data of 30 Hz 0 A is used as *D_s_*, while the real data are used as *D_t_*. *D_s_* in Table 1 and Table 2 both contain four health conditions: BR, CR, MI, and NO.

### 4.3. Result Comparison

To demonstrate the effectiveness of the proposed method, we compared it with ResNet [16], DeepCoral [21], DDC [22], and DANN [23]. To make an impartial comparison, all methods used the same ResNet network structure and parameters as the proposed method.

Figure 11 and Figure 12 illustrate the diagnostic accuracies of various methods. It can be observed that when both *D_s_* and *D_t_* are real data, the proposed method obtains a mean diagnostic precision of 98.02%. When *D_s_* is simulated data and *D_t_* is real data, as analyzed in Section 4.1, the transfer difficulty is significantly increased. Nevertheless, the proposed means still achieves a mean diagnostic precision of 83.83%, indicating its practical value. Furthermore, the proposed means outperforms other TL methods in all transfer tasks. This is because the proposed method introduces multiple domain classifiers and a weighted learning strategy, which enables the model to effectively measure the transferability of each label’s *D_s_* sample to *D_t_* and increase the contribution of shared label *D_s_* samples and decrease the contribution of outlier label *D_s_* samples during training. This effectively reduces the mismatching between *D_t_* samples and outlier label *D_s_* samples, thereby improving the diagnostic accuracy of transfer tasks.

### 4.4. Feature Visualization Analysis

To clearly demonstrate the feature distribution when simulation data is used as *D_s_* and real data as *D_t_*, the features extracted by various methods were visualized using t-SNE. Figure 13 depicts the feature visualization of different methods in task C5. In this task, there are four labels in *D_s_* samples, while there are only three labels in *D_t_* samples, and MI in *D_s_* belongs to the outlier label. From Figure 13, it can be seen that ResNet can accurately distinguish *D_s_* features of different labels, but the distribution of *D_t_* features it extracts is quite different from that of *D_s_* features, which leads to lower accuracy in classifying *D_t_* samples. In comparison, *D_s_* features and *D_t_* features extracted by DeepCoral, DDC, and DANN have a more similar distribution, but there is a large overlap between features of different labels, and some *D_t_* features are incorrectly aligned with *D_s_* MI features. In the proposed method, *D_t_* features can be correctly aligned with the corresponding *D_s_* features of the label, and the discriminability between features of different labels is higher, which further validates the effectiveness of the proposed means.

## 5. Conclusions

This paper puts forward a weighted domain adversarial neural network diagnostic model aimed at improving the fault diagnosis performance of PG in partial transfer tasks. Unlike traditional domain adaptation diagnostic methods that directly adapt all *D_s_* and *D_t_* class samples, this method considers the influence of outlier label *D_s_* samples. Specifically, this method uses multiple domain classifiers, each of which is responsible for matching samples of a certain label. And a weighting scheme is introduced to assign smaller weights to outlier label *D_s_* samples and domain classifiers responsible for matching outlier label source domain samples, effectively reducing the negative impact of outlier label *D_s_* samples and promoting correct matching of shared labeled *D_s_* samples and *D_t_* samples. When both *D_s_* and *D_t_* are real data, this means achieved an average diagnostic accuracy of 98.02%; when *D_s_* is simulated data and *D_t_* is real data, this method achieved an average diagnostic accuracy of 83.83%, both of which are better than other TL methods. In addition, this method relaxes the requirement that *D_s_* and *D_t_* need to have the same label space, which is more in line with practical application scenarios.

## Figures and Tables

**Figure 1 biomimetics-08-00361-f001:**
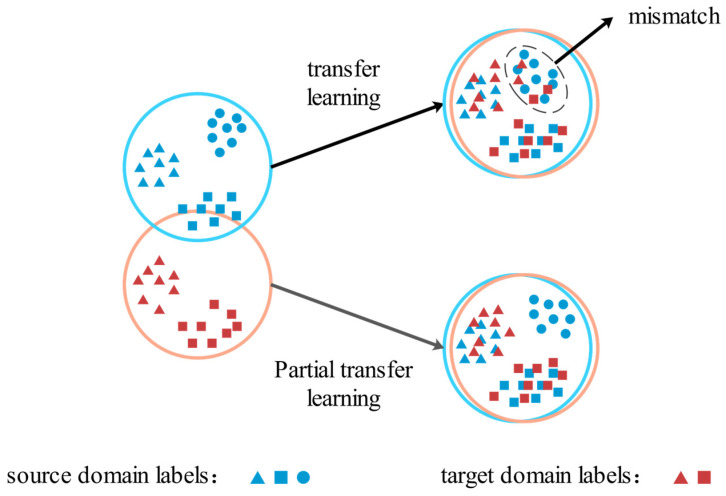
Schematic diagram of TL and PTL.

**Figure 2 biomimetics-08-00361-f002:**
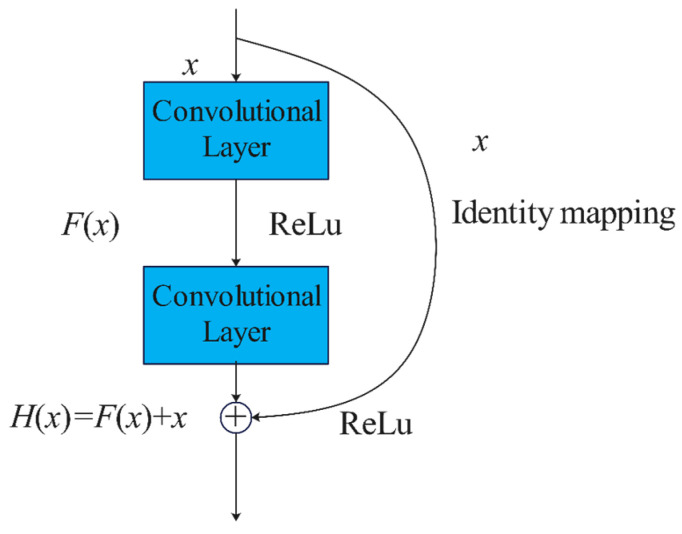
Schematic diagram of residual block.

**Figure 3 biomimetics-08-00361-f003:**
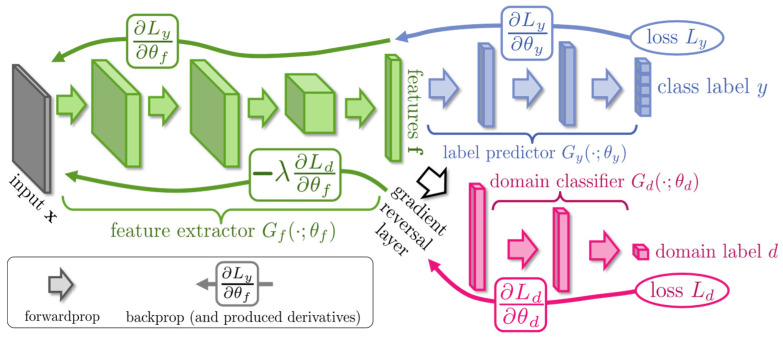
DANN framework.

**Figure 4 biomimetics-08-00361-f004:**
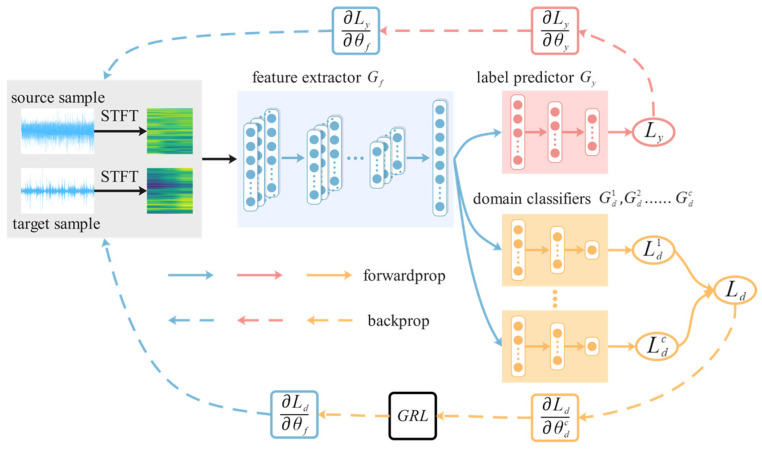
Weighted domain adversarial neural network framework.

**Figure 5 biomimetics-08-00361-f005:**
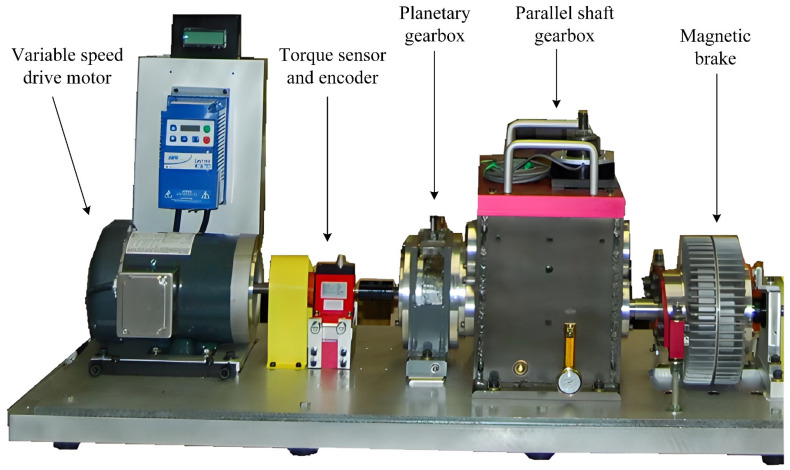
DDS experimental platform.

**Figure 6 biomimetics-08-00361-f006:**
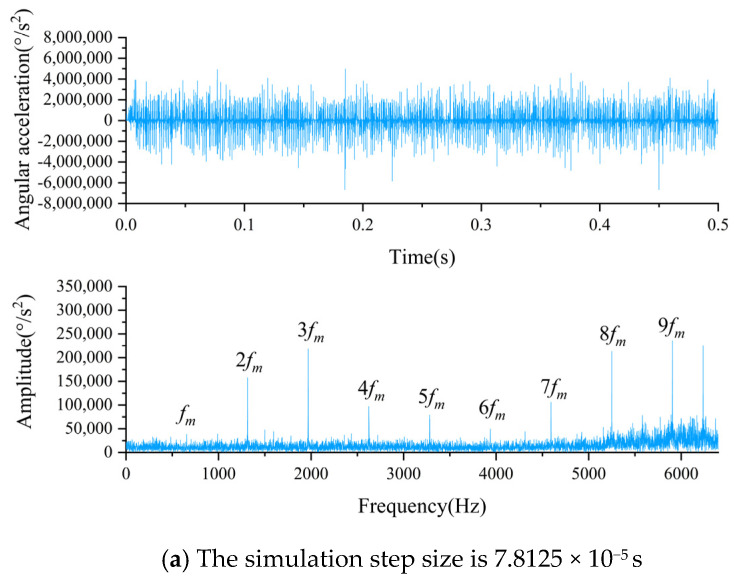
Time domain diagrams and frequency domain diagrams of different simulation step sizes of rigid-flexible coupling model.

**Figure 7 biomimetics-08-00361-f007:**
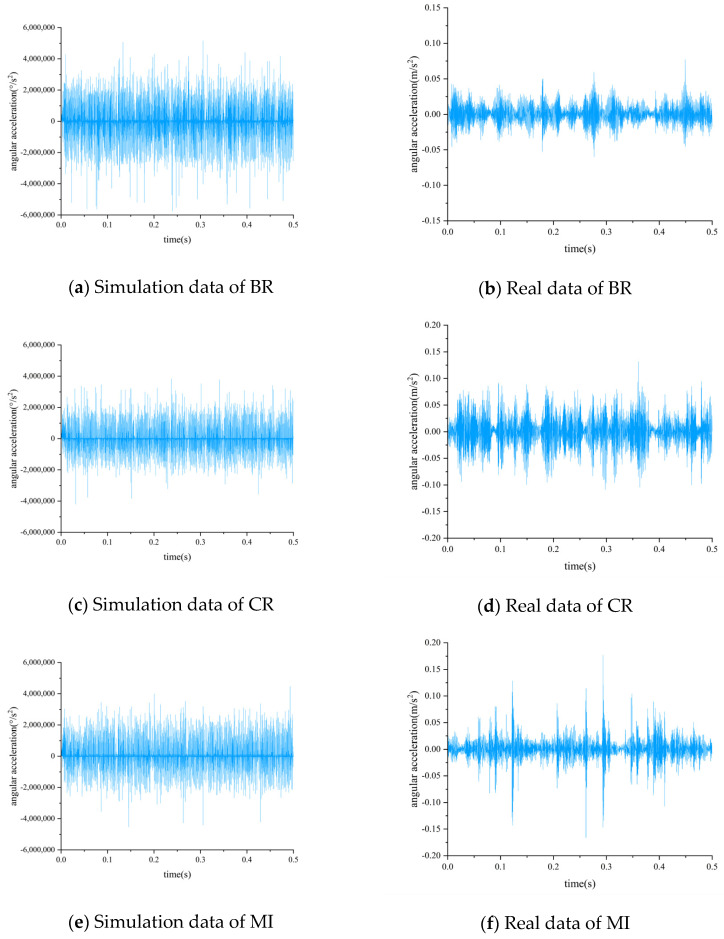
Comparison of simulation data and real data time-domain diagram.

**Figure 8 biomimetics-08-00361-f008:**
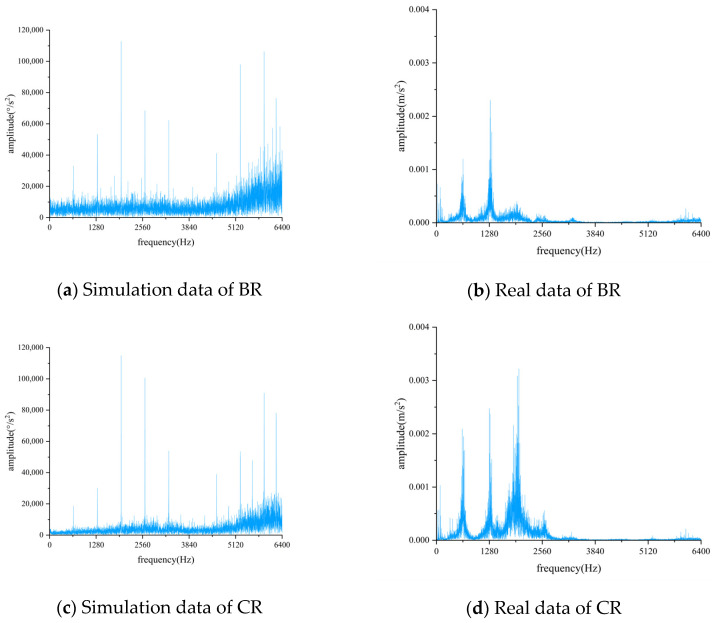
Comparison of simulation data and real data frequency domain diagram.

**Figure 9 biomimetics-08-00361-f009:**
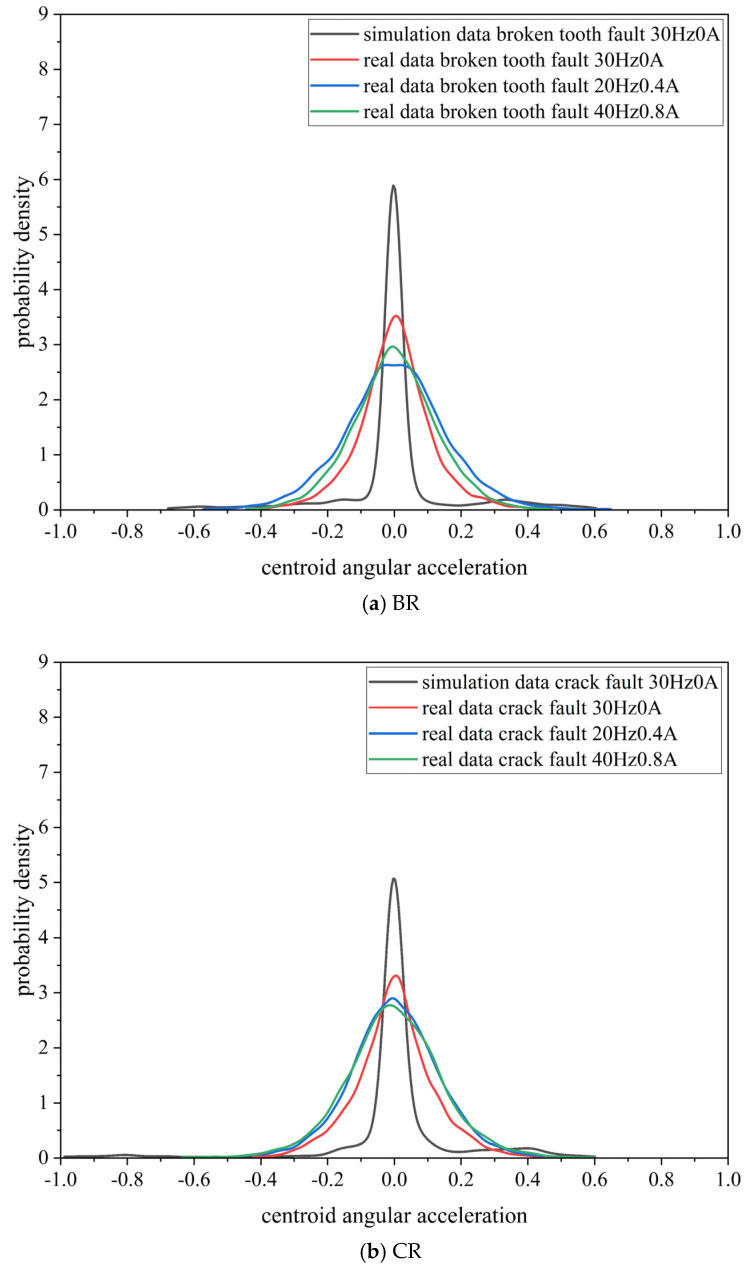
Probability distribution curves of simulated data and real data.

**Figure 10 biomimetics-08-00361-f010:**
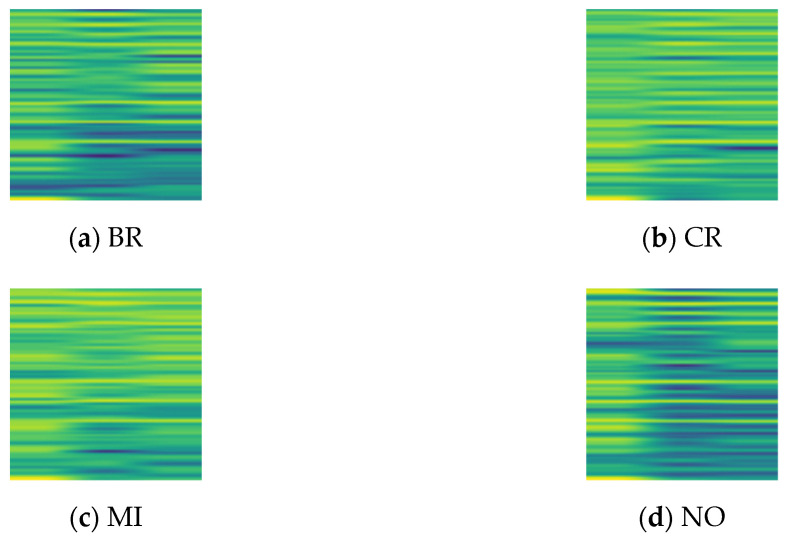
Time-frequency diagram of simulation data for different health conditions.

**Figure 11 biomimetics-08-00361-f011:**
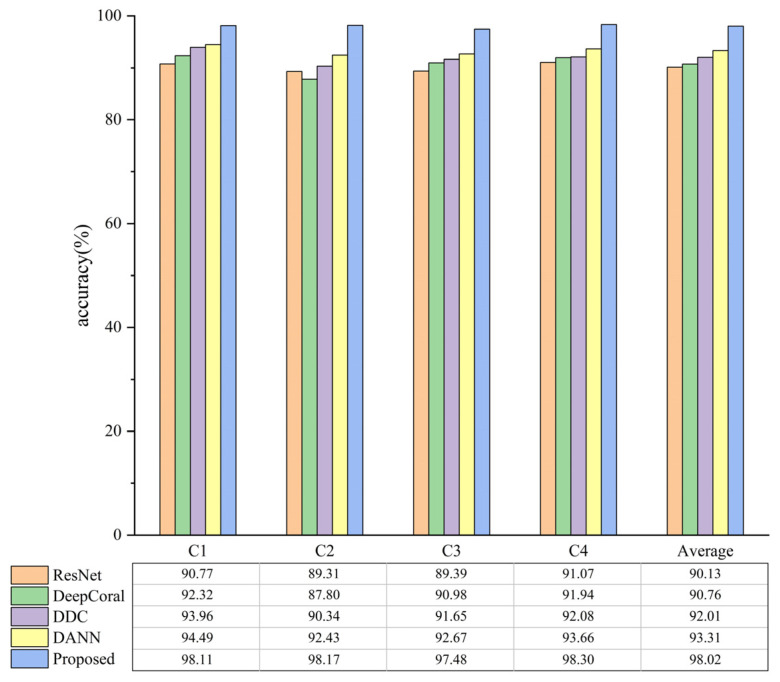
Partial transfer diagnostic accuracy with real data in both *D_s_* and *D_t_*.

**Figure 12 biomimetics-08-00361-f012:**
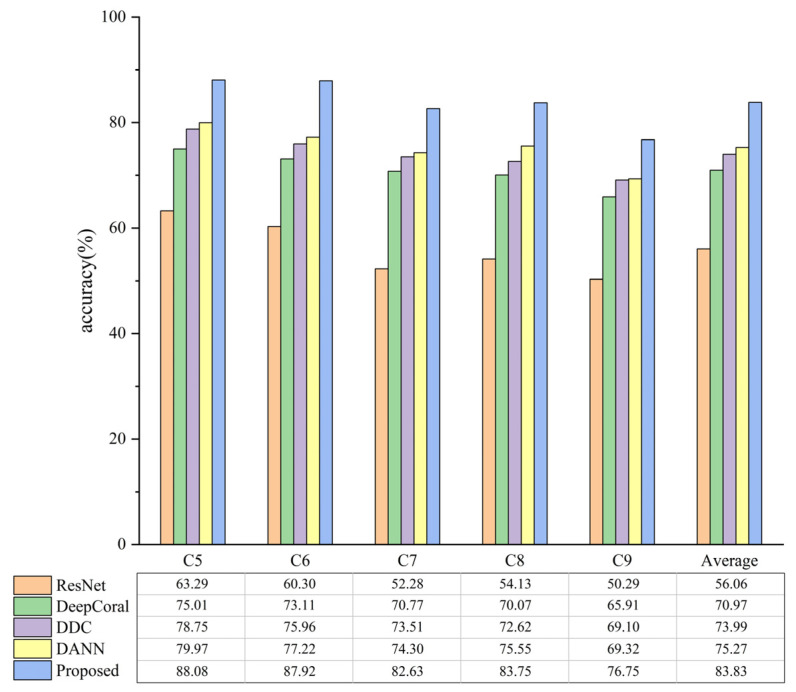
Partial transfer diagnostic accuracy with *D_s_* being simulated data and *D_t_* being real data.

**Figure 13 biomimetics-08-00361-f013:**
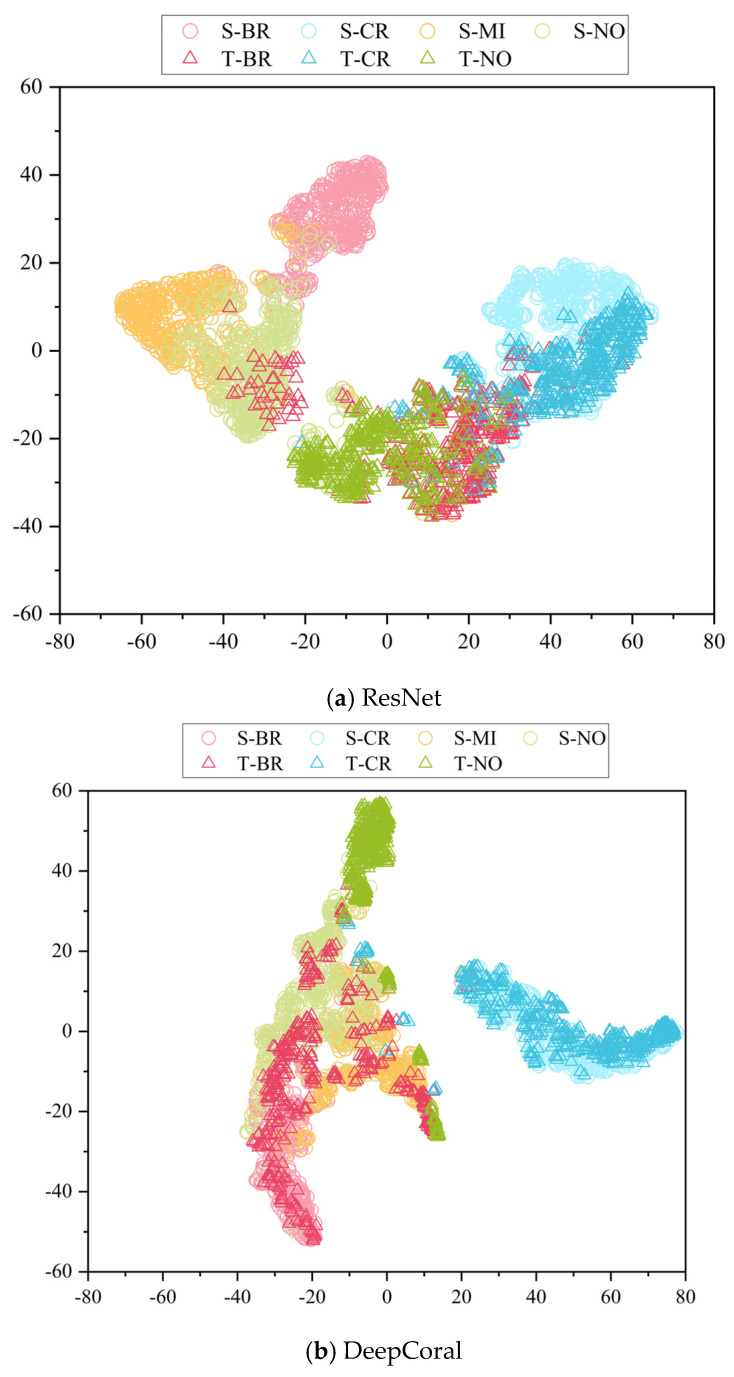
Feature visualization of different methods in partial transfer task C1.

**Table 1 biomimetics-08-00361-t001:** Partial transfer tasks where both *D_s_* and *D_t_* are real data.

Task Name	*D_t_* Conditions	*D_t_* Health Conditions
C_1_	30 Hz 0.8 A	BR, CR, MI
C_2_	20 Hz 0 A	BR, CR
C_3_	20 Hz 0.4 A	CR, NO
C_4_	40 Hz 0.8 A	CR

**Table 2 biomimetics-08-00361-t002:** Partial transfer tasks where *D_s_* is simulation data, and *D_t_* is real data.

Task Name	*D_t_* Conditions	*D_t_* Health Conditions
C_5_	30 Hz 0 A	BR, CR, NO
C_6_	30 Hz 0.8 A	BR, CR, MI
C_7_	20 Hz 0 A	BR, CR
C_8_	20 Hz 0.4 A	CR, NO
C_9_	40 Hz 0.8 A	CR

## Data Availability

The datasets generated during and/or analyzed during the current study are available from the corresponding author on reasonable request.

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
