# Peer review of "Fault Diagnosis of Planetary Gearbox Based on Dynamic Simulation and Partial Transfer Learning"

_biomimetics, 2023, doi:10.3390/biomimetics8040361_

Round 1
Reviewer 1 Report
The manuscript presents an interesting work on planetary gearbox diagnosis based on dynamic simulation and transfer learning techniques. The reviewer has several questions which need to be addressed by the authors. The questions are as follows.
1. The novel contributions of the research work are not clearly summarized in Section 1 of the manuscript.
2. As shown in Fig. 6, the differences between simulated data and real data are huge. How do the authors ensure that the simulated fault data of planetary gearboxes are of high-fidelity regarding real planetary gearboxes?
3. In lines 235-237 on page 7 of the manuscript, the authors stated that “During data acquisition, the input shaft speed was 30Hz with no load”. The scenario in which no load is considered is nonsense in terms of planetary gearbox fault diagnosis since no planetary gearbox works without load in the industry. The reviewer suggests deleting the “no load” case. Afterwards, the performance of the proposed method needs to be re-evaluated.
Manuscript writing looks good.
Author Response
Dear Editor and Reviewers,
Thank you for your letter and the reviewers’ comments concerning our manuscript entitled “Fault diagnosis of planetary gearbox based on dynamic simulation and partial transfer learning” (biomimetics-2532172). Those comments are all valuable and very helpful for revising and improving our paper, and also have the important guiding significance for our researches. We have studied comments carefully and have made correction which we hope meet with approval. Revised portion are marked in red in the paper.
Thanks very much for your attention to our paper.
Sincerely yours,
Meng-Meng Song, Zi-Cheng Xiong, Jian-Hua Zhong *, Shun-Gen Xiao *, Ji-Hua Ren

Reviewer 2 Report
Language and Style: The language used in the article is academic and technical, which is appropriate for a scientific research paper. The authors use precise and clear language to explain complex concepts, making it easier for readers familiar with the field to understand. The style is formal and consistent, which is typical for scientific literature. The use of figures and diagrams also helps to illustrate and support the points being made.
Logic and Structure: The paper appears to be well-structured, following a logical progression. It begins with an introduction, followed by a theoretical background section that explains relevant theories and concepts. The authors then describe the proposed method, followed by a practical case study. The paper concludes with a summary. This structure allows the authors to build their argument step by step, making it easier for readers to follow.
Importance and Quality: The paper seems to be of high quality, as it addresses a specific problem in the field of transfer learning and proposes a novel method to solve it. The authors support their claims with references to previous research, which adds credibility to their work. The inclusion of a practical case study also demonstrates the applicability of their method in real-world scenarios.
The paper presents a weighted domain adversarial neural network diagnostic model aimed at improving the fault diagnosis performance of partial transfer tasks. The authors have proposed a novel method that considers the influence of outlier label Ds samples, which is different from traditional domain adaptation diagnostic methods that directly adapt all Ds and Dt class samples.
The proposed method introduces multiple domain classifiers and a weighted learning strategy. This enables the model to effectively measure the transferability of each label's Ds sample to Dt and increase the contribution of shared label Ds samples while decreasing the contribution of outlier label Ds samples during training. This effectively reduces the mismatching between Dt samples and outlier label Ds samples, thereby improving the diagnostic accuracy of transfer tasks.
The authors have compared their method with other models like ResNet, DeepCoral, DDC, and DANN. All methods used the same ResNet network structure and parameters as the proposed method. The results show that the proposed method outperforms other transfer learning methods in all transfer tasks. When both Ds and Dt are real data, the proposed method obtains a mean diagnostic precision of 98.02%. When Ds is simulated data and Dt is real data, the proposed method still achieves a mean diagnostic precision of 83.83%, indicating its practical value.
The authors have also provided visualizations of the diagnostic accuracies of various methods, which further supports their claims. The paper concludes with a feature visualization analysis to clearly demonstrate the feature distribution when simulation data is used as Ds and real data as Dt.
The authors have acknowledged that their method relaxes the requirement that Ds and Dt need to have the same label space, which is more in line with practical application scenarios. This indicates that the authors have considered the practical implications and applicability of their method, which adds to the quality of the paper.
In conclusion, the paper presents a novel and effective method for improving the fault diagnosis performance of partial transfer tasks. The authors have provided a thorough explanation of their method, supported their claims with a comparison to other methods, and demonstrated the practical value of their method. This suggests that the paper is of high quality and could be of significant importance in the field of transfer learning.
Author Response
Dear Reviewer,
We truly appreciate the time and effort you dedicated to thoroughly reviewing our manuscript.
Your positive assessment of our paper has not only boosted our confidence but also validated the significance of our research. Your recognition serves as a motivation for us to continue exploring new avenues in our field of study.
Thank you for your time and support.
Sincerely yours,
Meng-Meng Song, Zi-Cheng Xiong, Jian-Hua Zhong *, Shun-Gen Xiao *, Ji-Hua Ren
Round 2
Reviewer 1 Report
The reviewer's questions have been addressed.
The Quality of English Language is good.